# 3D-GSRD: 3D Molecular Graph Auto-Encoder with Selective Re-mask Decoding

**Chang Wu**[1*], **Zhiyuan Liu**[2*], **Wen Shu**[3], **Liang Wang**[4], **Yanchen Luo**[1],
**Wenqiang Lei**[3], **Yatao Bian**[2], **Junfeng Fang**[2†], **Xiang Wang**[1†]

[1] University of Science and Technology of China [2] National University of Singapore
[3] Sichuan University [4] Institute of Automation, Chinese Academy of Sciences

wuchang0124@mail.ustc.edu.cn, acharkq@gmail.com, shuwen@stu.scu.edu.cn,
liang.wang@cripac.ia.ac.cn, luoyanchen@mail.ustc.edu.cn, wenqianglei@scu.edu.cn,
ybian@nus.edu.sg, fangjf1997@gmail.com, xiangwang1223@gmail.com

[*] Equal contribution.    [†] Corresponding author.

## Abstract

Masked graph modeling (MGM) is a promising approach for molecular representation learning (MRL). However, extending the success of re-mask decoding from 2D to 3D MGM is non-trivial, primarily due to two conflicting challenges: avoiding 2D structure leakage to the decoder, while still providing sufficient 2D context for reconstructing re-masked atoms. To address these challenges, we propose **3D-GSRD**: a **3D** Molecular **G**raph Auto-Encoder with **S**elective **R**e-mask **D**ecoding. The core innovation of 3D-GSRD lies in its **S**elective **R**e-mask **D**ecoding (**SRD**), which re-masks only 3D-relevant information from encoder representations while preserving the 2D graph structures. This SRD is synergistically integrated with a **3D R**elational-**Trans**former (**3D-ReTrans**) encoder alongside a structure-independent decoder. We analyze that SRD, combined with the structure-independent decoder, enhances the encoder's role in MRL. Extensive experiments show that 3D-GSRD achieves strong downstream performance, setting a new state-of-the-art on 7 out of 8 targets in the widely used MD17 molecular property prediction benchmark. The code is released at https://github.com/WuChang0124/3D-GSRD.

## 1 Introduction

Molecular representation learning (MRL) [1–3] is fundamental to a wide range of downstream tasks, including de novo drug design [4], molecular dynamics simulation [5], and molecular property prediction [6, 7]. Given the abundance of unlabeled molecular data in this field, self-supervised pretraining has emerged as a key strategy for learning effective molecular representations. Previous works have primarily focused on 1D molecular strings [8, 9] and 2D molecular graphs [10–12], achieving promising results. However, they often neglect critical 3D structural information, which is crucial for capturing molecular properties such as the highest occupied molecular orbital, molecular dynamics, and energy functions [13]. This limitation has led to a growing interest in incorporating 3D molecular coordinates into pretraining frameworks.

Masked graph modeling (MGM) has emerged as a leading paradigm for 3D molecular pretraining, aiming to learn data distributions by reconstructing randomly masked graph features [14, 15]. As illustrated in Figure 1, its 3D variant typically consists of three key components: (1) 3D graph masking, which perturbs the original 3D molecular graph by randomly masking features such as 3D coordinates, atom types, and chemical bonds [14, 16]; (2) a 3D graph encoder, which processes the

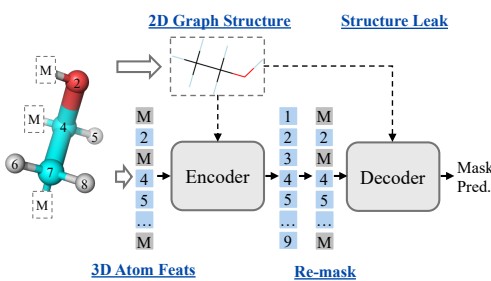

Figure 1: Illustration of 3D MGM with re-mask decoding and 2D structure leakage.

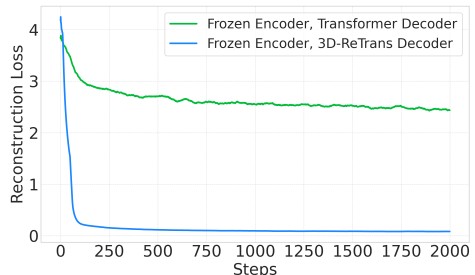

Figure 2: Reconstruction loss across two pre-training settings. We compare settings using frozen encoders (3D-ReTrans) and structure-independent (Transformer) versus structure-dependent (3D-ReTrans) decoders.

masked graph to generate molecular representations; and (3) a 3D graph decoder, which reconstructs the masked features from the encoded representations. After pretraining via reconstruction, the 3D graph encoder is finetuned on downstream tasks to enhance performance.

However, a long-standing challenge in MGM, mirroring similar issues in masked image modeling [17], is the misalignment between the reconstruction objective and representation learning [18]. Specifically, minimizing the reconstruction error often leads models to focus on low-level graph features, such as atom types and 3D coordinates, rather than learning high-level graph semantics required for property prediction [19]. To mitigate this, re-mask decoding is introduced for MGM pretraining on 2D molecular graphs [15, 20, 18]. This method re-masks the encoder's representation of previously masked atoms before feeding them to the decoder (Figure 1). In this way, the encoder is prevented from reconstructing the masked atoms directly and is encouraged to focus on generating high-quality representations of the unmasked graph regions, which the decoder then uses for reconstruction. This approach shifts the encoder's focus from graph reconstruction to MRL and yields substantial downstream improvements for 2D MGM.

To adapt re-mask decoding for 3D MGM, we identify two seemingly contradictory challenges:

- **Leaking 2D structure to decoder weakens encoder's MRL capability.** The decoder should rely solely on the encoder representations to reconstruct masked features. Exposing the decoder directly to 2D molecular structures (*e.g.,* chemical bond connections) diminishes the encoder's role in MRL, as the decoder can recover masked features using the provided 2D molecular structures, even with poor encoder representations. For example, Figure 2 shows that a frozen randomly-initialized encoder with a trainable structure-dependent decoder can achieve relatively low reconstruction loss when predicting masked atomic coordinates. Consequently, the encoder focuses less on capturing structures, leading to suboptimal MRL performance. However, existing re-mask decoding methods [15, 18, 20] typically use structure-dependent decoders like graph neural networks [21], which exacerbate this issue.

- **Structure-independent decoding can prevent structure leakage, but hinders reconstruction of re-masked atoms.** A naive solution to prevent structure leakage is to use a structure-independent decoder, which consumes no 2D structure input. However, this approach fails to account for the relative positions and contextual relationships of re-masked atoms within the 2D graph, making it challenging to distinguish between re-masked atoms during reconstruction. To address this, we propose leveraging the 3D graph encoder to generate 2D structural contexts for re-masked atoms. This ensures that the encoder is effectively trained for structural representation while preventing structure leakage beyond the encoder's representations.

To address the challenges above, we introduce **3D** Molecular **G**raph Auto-Encoder with **S**elective **R**e-mask **D**ecoding (**3D-GSRD**), a 3D MGM framework with three key elements: (1) the **S**elective **R**e-mask **D**ecoding (**SRD**) that re-masks only 3D-relevant information from the encoder representations while preserving its 2D structural context; (2) a structure-independent decoder that derives all structural information exclusively from the encoder; and (3) **3D Re**lational-**Trans**former (termed as **3D-ReTrans**) as 3D graph encoder, effectively integrating 3D molecular features (*e.g.,* atomic coordinates) and 2D features (*e.g.,* bonding connections) for MRL.

Specifically, SRD ensures the preservation of 2D graph structures during re-masking by reintroducing 2D information through a **2D** graph **P**osition **E**ncoder (**2D-PE**). Crucially, this 2D-PE is trained via distillation from the 3D graph encoder's representations, ensuring its information is fully contained within the 3D encoder, as demonstrated in Section 5. The distillation process also allows the 2D-PE's structural encoding capability to improve alongside the 3D encoder's advancements during pretraining. To complement SRD, we employ a structure-independent Transformer [22] decoder that derives 2D graph structure exclusively from the 3D encoder and the 2D-PE. Together with SRD, our decoder provides the re-masked atoms with rich 2D graph contexts that are distilled from the 3D graph encoder, while preventing 2D structure leakage.

Encoding 3D molecules is challenging due to their multi-modal nature (*e.g.,* discrete atom types vs. continuous coordinates) and multi-granular structure (*e.g.,* atom-wise vs. pair-wise features). Prior works like PaiNN [23] and TorchMD-NET [24] address this by using equivariant architectures that separately process scalar features (*e.g.,* atom types, distances) and vector features (*e.g.,* directional geometry). Building on these insights, we introduce **3D-ReTrans** as our 3D graph encoder, extending the Relational-Transformer [25] to incorporate both scalar and vector features while maintaining its scalability and flexibility to process both atom-wise and pair-wise features. Specifically, we introduce a tailored attention mechanism that incorporates pairwise distances and interactions directly into attention weights, along with a 3D Update Layer that jointly updates scalar and vector features. This design yields strong performance on MRL and serves as a robust backbone for 3D MGM pretraining.

Finally, we include in-depth analysis to showcase the inner mechanism of SRD and our structure-independent decoder, demonstrating our key claims of shifting the encoder's focus to MRL while preventing structure leakage in the decoder. Based on these revealed advantages in MGM pretraining, 3D-GSRD demonstrates superior performance when being fine-tuned on downstream datasets, achieving new state-of-the-art on 7 out of 8 molecules for MD17 [26].

## 2   Related Work

Molecular pretraining has emerged as a fundamental approach for molecular representation learning [27–29], critical for various downstream tasks, such as molecule property prediction.

**3D Molecular Denoising and Masked Graph Modeling.** Recent advances in 3D molecular pretraining have focused on 3D structure learning through coordinate denoising and masking. For example, [30–32] introduce noise to atomic coordinates and then reconstruct them. SubGDiff [2] adds distinct Gaussian noise to different substructures of 3D molecular conformation and performs denoising via a diffusion process. MolSpectra [33] uses the energy spectra to enhance 3D molecular representation learning during denoising. As for masking, Uni-Mol [14] and Uni-Mol2 [16] employ masked coordinates prediction as one of the self-supervised tasks, while other works like [1] focus on masking and predicting bond lengths and angles.

**Other 3D Molecular Pretraining Methods.** EPT [34] proposes a multi-domain 3D pretraining approach by combining atom-level features for small molecules and residue-level features in proteins. 3D PGT [35] designs three generative pretraining tasks, including predicting bond length, bond angle, and dihedral angle, and introduces an adaptive fusion strategy for these tasks, using total energy as a surrogate metric to optimize their combination weight. GraphMVP [36] and 3D Infomax [7] use contrastive learning to transfer knowledge from the 3D encoder into the 2D graph encoder.

**Graph Position Encoding and Structure Encoding.** Position Encoding (PE) encodes the spatial position of a given node within a graph [37]. Some methods [38, 39] use adjacency, Laplacian, or distance matrices to represent PE. Other approaches like [40–42] leverage shortest paths, heat kernels, or Green's function to compute pair-wise distance, capturing the distance and directional relationships between nodes. Currently, in MOL-AE [19], SMILES strings are used to provide PE to the decoder as an identifier. Structure Encoding (SE) encodes the structural information of graphs and subgraphs. Common methods include node degree [43], Laplacian matrices [38], and Boolean indicators that specify whether two nodes belong to the same substructure [44]. Unlike these methods, we propose using a 2D graph position encoder distilled from a 3D graph encoder to produce SE, which provides rich and effective context information.

**2D Molecular Graph Pretraining.** Previous methods primarily focus on leveraging 2D molecular graphs to learn molecular representation. A popular technique is masked graph modeling [15, 8, 10],

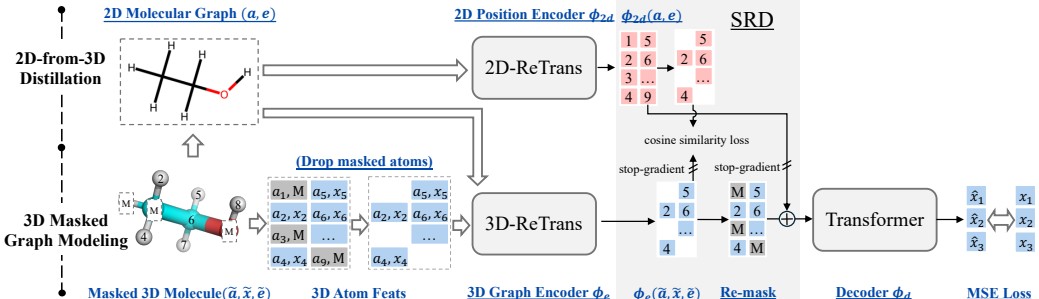

Figure 3: Overview of 3D-GSRD. It contains three key elements: (1) a 3D-ReTrans encoder; (2) the SRD that re-masks only 3D-relevant information from the encoder representations while preserving its 2D structure information via 2D-from-3D distillation; (3) a structure-independent decoder.

typically comprising three key components [18]: graph tokenizer [11, 3], graph masking [45, 46], and graph autoencoder [47–49]. Another prominent line of work is contrastive learning [50, 12, 51], which aims to pull positive pairs and push negative pairs apart in the representation space. Notably, methods such as GeomGCL [6], GraphMVP [36], and 3D Infomax [7] incorporate 3D molecular conformations as auxiliary information to enhance 2D graph representations via contrastive objectives. While effective for 2D molecular pretraining, these methods overlook 3D features, which are crucial for molecular representation learning. Moreover, directly extending these methods to 3D molecular pretraining is non-trivial due to the increased complexity and spatial nature of 3D molecular data.

Our method is similar to GraphMVP [36] in leveraging both 2D and 3D molecular graphs but differs in objective and design. While GraphMVP transfers 3D information into a 2D encoder to enhance 2D graph representations, our method distills 3D representation into 2D-PE, ensuring the 2D-PE's embedding is fully contained within the 3D encoder to avoid structure leakage in decoding. Additionally, GraphMVP aligns the 2D and 3D views of the same molecule and contrasts views of different molecules using contrastive losses. Our method instead uses a cosine similarity loss that encourages 2D-PE to generate structural encodings closely aligned with the 3D encoder. This offers a simpler and more efficient framework for 2D structure-informed decoding without structure leakage.

## 3 Preliminary: 3D Masked Graph Modeling

**Notations.** A 3D molecular graph is represented as $G = (\mathbf{x}, \mathbf{a}, \mathbf{e})$, where $\mathbf{x} \in \mathbb{R}^{N \times 3}$ denotes the 3D atomic coordinates, $\mathbf{a} \in \mathbb{R}^{N \times *}$ represents the atom types, and $\mathbf{e} \in \mathbb{R}^{N \times N \times *}$ captures the atomic pair features, such as inter-atomic distances and bonds. $N$ is the number of atoms.

**Graph Masking.** Given a molecular graph $G$, the 3D coordinates $\{\mathbf{x}_i \in \mathbb{R}^3 | i \in \mathcal{V}_m\}$ of a randomly selected subset of atoms $\mathcal{V}_m$ are masked. For each masked atom $i \in \mathcal{V}_m$, its original coordinates $\mathbf{x}_i$ is replaced by a learnable special token $\mathbf{m}_x \in \mathbb{R}^3$. The coordinate matrix $\mathbf{x}$ after masking is denoted as $\tilde{\mathbf{x}}$, and the masked graph is denoted as $\tilde{G} = (\tilde{\mathbf{x}}, \mathbf{a}, \mathbf{e})$. Some prior works [19] instead remove all information corresponding to the masked atoms, including their 3D coordinates, atom types, and pairwise features, which results in a masked graph $\tilde{G} = (\tilde{\mathbf{x}}, \tilde{\mathbf{a}}, \tilde{\mathbf{e}})$. In this work, we adopt this latter masking strategy, fully excluding the masked atoms from the input graph.

**3D Graph Auto-Encoder and 2D Structure Leakage.** The 3D graph auto-encoder comprises a graph encoder $\phi_e(\cdot)$ and a graph decoder $\phi_d(\cdot)$. The encoder processes the masked graph $\tilde{G}$ to produce graph representations $\mathbf{h} = \phi_e(\tilde{G}) \in \mathbb{R}^{N \times *}$. The decoder then predicts the masked coordinates $\{\hat{\mathbf{x}}_i | i \in \mathcal{V}_m\}$ using $\phi_d(\mathbf{h})$, and optionally incorporating the pair features $\phi_d(\mathbf{h}, \mathbf{e})$. However, using pair features $\mathbf{e}$ introduces **2D structure leakage**, as the decoder relies on additional information beyond the encoder's representation $\mathbf{h}$. This weakens the encoder's role in MRL, because the decoder can leverage pair features to compensate for any deficiencies in the encoder's representations. Despite this drawback, such leakage is common in previous MGM works [15, 20, 18] that utilize Graph Neural Networks [21] as decoders. In contrast, methods that avoid 2D structure leakage [14, 16] mostly use weak decoders, such as MLPs, which can lead to suboptimal MGM pretraining.

**Re-mask Decoding.** Before passing $\mathbf{h}$ into the decoder, re-mask decoding replaces the representations of the previously masked atoms $\mathcal{V}_m$ with a learnable token $\mathbf{m}_h$, preventing the encoder from directly

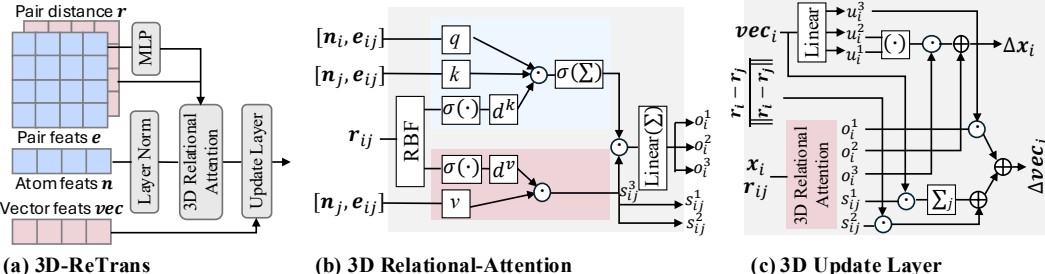

**(a) 3D-ReTrans**    **(b) 3D Relational-Attention**    **(c) 3D Update Layer**

Figure 4: Illustration of 3D-ReTrans. **(a)** 3D-ReTrans is constructed by stacking multiple 3D Relational-Attention and 3D Update Layers. **(b)** 3D Relational-Attention that processes both atom-wise and pair-wise features. **(c)** 3D Update Layer that includes a residual connection.

predicting the masked coordinates. This encourages the encoder to focus on learning meaningful representations for the unmasked graph regions. The re-masked representation $\tilde{\mathbf{h}}$ is defined as:

$$\tilde{\mathbf{h}}_i = \text{re-mask}(\mathbf{h}_i) = \begin{cases} \mathbf{m}_h, & \forall i \in \mathcal{V}_m, \\ \mathbf{h}_i, & \text{otherwise.} \end{cases} \tag{1}$$

**MGM Loss.** The pretraining objective minimizes the mean squared error between the ground truth coordinates $\{\mathbf{x}_i | i \in \mathcal{V}_m\}$ of the masked atoms and the decoder's predicted coordinates $\{\hat{\mathbf{x}}_i | i \in \mathcal{V}_m\}$:

$$\mathcal{L}_{\text{MGM}} = \sum_{i \in \mathcal{V}_m} \|\hat{\mathbf{x}}_i - \mathbf{x}_i\|^2. \tag{2}$$

## 4 Methodology: 3D-GSRD

In this section, we present our method **3D** Molecular **G**raph Auto-Encoder with **S**elective **R**e-mask **D**ecoding (**3D-GSRD**) (Figure 3). Below, we start by elaborating on the Selective Re-mask Decoding and the pretraining objective of 3D-GSRD. We then describe the encoder of 3D-ReTrans.

### 4.1 SRD: Selective Re-mask Decoding

Here we introduce SRD to improve 3D MGM. Re-mask decoding is proposed to address the mismatch between the reconstruction objective of 2D MGM and MRL [15, 18]. Our SRD extends this approach to 3D MGM while overcoming issues such as 2D structure leakage and providing 2D contexts to re-masked atoms.

**Re-mask Decoding with 2D Graph Position Encoder**. As Figure 3 shows, given the encoder representation $\mathbf{h} = \phi_e(\tilde{\mathbf{x}}, \tilde{\mathbf{a}}, \tilde{\mathbf{e}})$ of $\tilde{G}$, SRD can be defined as:

$$\text{SRD}(\mathbf{h}, \tilde{G}) = \text{re-mask}(\mathbf{h}) + \text{stop-grad}(\phi_{2d}(\mathbf{a}, \mathbf{e})), \tag{3}$$

where $\text{SRD}(\mathbf{h}, \tilde{G})$ is directly fed into the decoder for masked prediction; re-mask$(\cdot)$ is the standard re-mask; and stop-grad$(\cdot)$ stops the gradient flow to the 2D graph position encoder $\phi_{2d}$, which generates $\tilde{G}$'s 2D representation $\phi_{2d}(\mathbf{a}, \mathbf{e}) \in \mathbb{R}^{N \times *}$.

**Building a 2D Graph Position Encoder without Structure Leakage via 2D-from-3D Distillation.** The 2D graph position encoder $\phi_{2d}(\mathbf{a}, \mathbf{e})$ is the key component of SRD. For unmasked atoms, $\phi_{2d}$ conveys the same information as the 3D graph encoder $\phi_e$, preventing any information leakage beyond what $\phi_e$ has captured. For re-masked atoms, $\phi_{2d}$ offers the necessary 2D contexts that would have been available from $\phi_e$ without re-masking, enabling the decoder to distinguish the relative positions of re-masked atoms. To prevent information leakage, $\phi_{2d}$ is trained exclusively through knowledge distillation from the 3D encoder, without any gradient updates from the MGM loss. This is enforced by the stop-grad$(\cdot)$, which blocks the gradient flow from the MGM loss into $\phi_{2d}$. Further, the knowledge distillation loss for $\phi_{2d}$ can be written as:

$$\mathcal{L}_{\text{distill}} = -\sum_{i \notin \mathcal{V}_m} \cos(\phi_{2d}(\mathbf{a}, \mathbf{e})_i, \text{stop-grad}(\phi_e(\mathbf{x}, \mathbf{a}, \mathbf{e})_i)), \tag{4}$$

where $\cos(\cdot, \cdot)$ denotes cosine similarity. The loss applies only to unmasked atoms for the consistency of $\phi_{2d}$'s training objective. We employ stop-grad$(\cdot)$ to prevent updating the 3D graph encoder $\phi_e$, allowing it to focus on MRL.

While the 2D graph position encoder can be any graph encoder, we implement it as a 2D-ReTrans, a simplified version of the 3D-ReTrans that excludes 3D coordinates and distance inputs. Our experiments demonstrate the effectiveness of this design.

### 4.2 Pretraining 3D-GSRD

**3D Graph Auto-Encoder.** We employ the 3D-ReTrans as encoder and employ SRD with a structure-independent decoder of transformer [22]. In this way, we avoid structure leakage to the decoder beyond the encoder's information and provide strong 2D structural contexts for the re-masked atoms.

**Pretraining Loss.** We combine the MGM loss and 2D-from-3D distillation loss for pretraining:
$$\mathcal{L}_{\text{pretrain}} = \mathcal{L}_{\text{MGM}} + \mathcal{L}_{\text{distill}}. \tag{5}$$

### 4.3 3D Relational-Transformer

Encoding 3D molecular graphs $G = (\mathbf{x}, \mathbf{a}, \mathbf{e})$ presents significant challenges of processing 3D coordinates $\mathbf{x}$ while preserving 3D equivariance and integrating the pairwise features $\mathbf{e}$, whose shape $(N, N, *)$ differs from the atomic features $(N, *)$. Prior works have primarily focused on ensuring 3D equivariance while processing the 3D coordinates $\mathbf{x}$ [24, 52], but have paid less attention to effectively incorporating pair features $\mathbf{e}$. Most existing methods [14, 52, 53] incorporate pairwise representations of scalar values in self-attention layers [22], limiting their ability to capture the high-dimensional nature of inter-atomic interactions. While TorchMD-NET [24] models distances as high-dimensional pairwise features, extending it to include chemical bonds remains challenging.

To address the challenges, we propose **3D-ReTrans** as our encoder, leveraging the Relational-Transformer's [25] scalability and flexibility to incorporate pair features, while enabling it to process 3D coordinates. Draw inspiration from prior works [24, 23, 54], a core design is to explicitly separate and jointly process two types of features: (1) scalar features, which encode scalar information like atom types and distances; (2) vector features, which capture directional geometric information. Based on this, our key enhancements focus on improving its attention mechanism and incorporating a 3D update layer. More details about 3D-ReTrans are provided in Appendix B.

**3D Relational-Attention.** Each atom is represented by concatenating its types and coordinates: $\mathbf{n}_i = [\mathbf{a}_i; \mathbf{x}_i]$. The interaction between atoms $i$ and $j$ is captured by the pair feature $\mathbf{e}_{ij} \in \mathbb{R}^d$ and their Euclidean distance $r_{ij}$. 3D Relational-Attention is defined as:

$$\mathbf{q}_{ij} = [\mathbf{n}_i, \mathbf{e}_{ij}]\mathbf{W}^q, \tag{6} \qquad [\mathbf{k}_{ij}; \mathbf{v}_{ij}] = [\mathbf{n}_j; \mathbf{e}_{ij}][\mathbf{W}^k; \mathbf{W}^v], \tag{7}$$

$$[\mathbf{d}_{ij}^k; \mathbf{d}_{ij}^v] = \text{SiLU}\left([\mathbf{W}^{dk}; \mathbf{W}^{dv}]e^{\text{RBF}}(r_{ij})\right), \tag{8} \qquad [\mathbf{s}_{ij}^1, \mathbf{s}_{ij}^2, \mathbf{s}_{ij}^3] = \mathbf{v}_{ij} \odot \mathbf{d}_{ij}^v, \tag{9}$$

$$\boldsymbol{\alpha}_{ij} = \text{SiLU}_j\left(\frac{\mathbf{q}_{ij} \cdot (\mathbf{k}_{ij} \odot \mathbf{d}_{ij}^k)}{\sqrt{d}}\right), \tag{10} \qquad [\mathbf{o}_i^1, \mathbf{o}_i^2, \mathbf{o}_i^3] = \mathbf{W}^f(\sum_{j=1}^N \boldsymbol{\alpha}_{ij}\mathbf{s}_{ij}^3), \tag{11}$$

where $e^{\text{RBF}}(\cdot) : [0, \infty) \to \mathbb{R}^d$ is a distance expansion function to encode the distance variable into a $d$-dimensional vector [24] (see Appendix B). The terms $\mathbf{W}^q$, $\mathbf{W}^k$, $\mathbf{W}^v$, $\mathbf{W}^{dk}$, $\mathbf{W}^{dv}$, and $\mathbf{W}^f$ are learnable linear projectors, and $\odot$ denotes element-wise product. The output scalar features $\mathbf{o}_i^1$, $\mathbf{o}_i^2$, and $\mathbf{o}_i^3$ encode pairwise interactions and interatomic distances. Moreover, the attention mechanism facilitates the integration of distance information into the vector features via scalar filters $\mathbf{s}_{ij}^1$ and $\mathbf{s}_{ij}^2$ within the subsequent 3D Update Layer.

**3D Update Layer.** The 3D Update Layer facilitates information exchange between scalar and vector features. Vector features $\mathbf{vec}_i \in \mathbb{R}^{* \times 3}$ are initialized as zeros and jointly updated with scalar features $x_i$. The update $\Delta x_i$ and $\Delta \mathbf{vec}_i$ are defined as:

$$[\mathbf{u}_i^1, \mathbf{u}_i^2, \mathbf{u}_i^3] = \mathbf{W}^v(\mathbf{vec}_i), \tag{12} \qquad \mathbf{w}_j = \sum_{j=1}^N \left(\mathbf{vec}_j \odot \mathbf{s}_{ij}^1\right) + \mathbf{s}_{ij}^2 \odot \frac{\mathbf{r}_i - \mathbf{r}_j}{\|\mathbf{r}_i - \mathbf{r}_j\|}, \tag{13}$$

$$\Delta x_i = \mathbf{o}_i^2 + \mathbf{o}_i^3 \odot (\mathbf{u}_i^1 \cdot \mathbf{u}_i^2), \tag{14} \qquad \Delta \mathbf{vec}_i = \mathbf{u}_i^3 \odot \mathbf{o}_i^1 + \mathbf{w}_j, \tag{15}$$

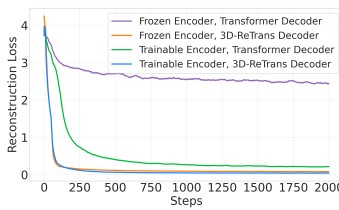 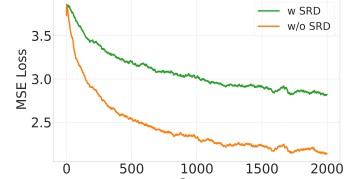 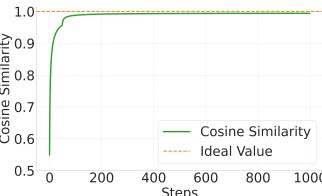

Figure 5: Reconstruction loss across four pretraining settings.

Figure 6: Probe encoder for masked atom coordinates when pretrained with/without SRD.

Figure 7: Reconstructing 2D-PE representation using 3D encoder representation.

where $\mathbf{W}^v$ are learnable linear projectors. Scalar features incorporate vector information via element-wise multiplication with the scalar product of vector components, while vector features are updated using both directional features $\mathbf{w}_j$ and a scalar filter $\mathbf{o}_i^1$.

As shown in Figure 4, the 3D-ReTrans is constructed by stacking multiple 3D Relational-Attention and 3D Update Layers. Each layer performs residual updates on both scalar features $\Delta x_i$ and vector features $\Delta \mathbf{vec}_i$, allowing the model to simultaneously capture scalar properties (*e.g.,* atom types, distances) and expressive directional geometric information. For pretraining, the final outputs $x_i$ and $\mathbf{vec}_i$ are fed into the decoder, while for finetuning, they are passed to the prediction head.

**Learning 3D Equivariance and Invariance by Data Augmentations.** Considering that our model lacks built-in 3D equivariance or invariance, we leverage data augmentations to instill these symmetries, following AlphaFold3 [53]. During MGM pretraining, atomic coordinates $\mathbf{x}$ are randomly rotated using transformations sampled from the SO(3) group and translated with offsets drawn from $\mathbf{t} \sim \mathcal{N}(\mathbf{0}, 0.01\mathbf{I}_3)$. These augmentations encourage the model to adjust its predictions equivariantly with any rotations and small translations. During fine-tuning for property prediction, the same augmentations are applied, but the model is trained to predict consistent properties, thereby learning invariance to rotations and translations.

We favor data-augmented equivariance with relational attention over fully E(3)-equivariant message passing. While built-in equivariant architectures offer formal guarantees, they often restrict how pair features are parameterized and incur non-trivial computational overhead (*e.g.,* tensor bases, spherical harmonics), which can hinder scaling and complicate integration with diverse molecular cues. In contrast, our approach instills rotational and translation robustness through augmentations, allowing encoder to operate with lightweight vector and scalar updates and to flexibly ingest high-dimensional pair features without architectural surgery. This yields a plug-and-play backbone that is easier to optimize, accommodates sparsity and density variations, and remains representation-rich: relational attention can expand or swap pair features as downstream tasks evolve, while maintaining competitive robustness to pose changes at a substantially lower training and inference cost.

# 5 Analyzing Selective Re-mask Decoding and Structure-Independent Decoder

In this section, we conduct extensive experiments to evaluate the components of 3D-GSRD, focusing on the structure-independent decoder and SRD, including its key part 2D-from-3D distillation. We analyze their effects on the overall performance of the 3D-GSRD framework and how they contribute to 3D MGM pretraining.

**Analysis 1. The structure-dependent decoder can diminish the encoder's role in MRL.** We pretrain the auto-encoder framework under four settings, all using 3D-ReTrans as the encoder: (1) a frozen encoder with a Transformer decoder; (2) a frozen encoder with a 3D-ReTrans decoder; (3) a trainable encoder with a Transformer decoder; and (4) a trainable encoder with a 3D-ReTrans decoder. Figure 5 reports the reconstruction loss of masked atom coordinates, averaged over every 50 batches. When the encoder is frozen, the Transformer decoder (*i.e.,* structure-independent decoder) struggles to reconstruct masked atoms coordinates, yielding high reconstruction loss due to poor input representations and the absence of 2D molecular structural information. In contrast, with the 3D-ReTrans decoder (*i.e.,* structure-dependent decoder), which leverages 2D molecular structures as input, the loss decreases rapidly during pretraining, even with a frozen encoder. This demonstrates that a powerful, structure-dependent decoder can compensate for weak encoder representations, diminishing the encoder's role in MRL.

**The structure-independent decoder heavily relies on high-quality encoder representations.** When paired with a trainable encoder, the structure-independent decoder achieves a much lower reconstruction loss, indicating that it relies heavily on the encoder to provide informative representations. In contrast, with a structure-dependent decoder, the loss remains low regardless of the encoder representation's quality, highlighting that such decoders reduce the learning pressure on the encoder and can hinder its ability to learn meaningful molecular features.

**Analysis 2. Structure-independent decoder improves downstream performance compared to structure-dependent decoder.** We pretrain the auto-encoder using either a structure-dependent or a structure-independent decoder while keeping all other settings constant, then finetune the pretrained encoder on molecular property prediction tasks. As Table 1 shows, the structure-independent decoder achieves better performance, outperforming the structure-dependent decoder by 5% in Toluene. This demonstrates that using a structure-independent decoder encourages the encoder to learn more informative representations, leading to improved performance on downstream tasks.

**Analysis 3. 2D-PE and 2D-from-3D distillation boost downstream performance.** We pretrain the auto-encoder with and without 2D-PE and 2D-from-3D distillation, followed by finetuning the encoder on MD17 datasets. As shown in Table 1, combining 2D-PE and distillation consistently improves performance. In contrast, using 2D-PE alone leads to degradation, likely due to unintended leakage of 2D structural information into the decoder. Moreover, 2D-from-3D distillation guides the 2D-PE to focus on encoding

Table 1: Analyzing the decoder and SRD. Performance (MAE $\downarrow$) on MD17. The variant without SRD and $\mathcal{L}_{distill}$ corresponds to the model ablated without 2D-PE.

| Decoder | SRD | $\mathcal{L}_{distill}$ | Salicylic | Toluene | Uracil |
|---|---|---|---|---|---|
| Structure-dependent | ✓ | ✓ | 0.0401 | 0.0291 | 0.0334 |
| Structure-independent | ✗ | ✗ | 0.0404 | 0.0292 | 0.0328 |
| Structure-independent | ✓ | ✗ | 0.0416 | 0.0293 | 0.0329 |
| Structure-independent | ✓ | ✓ | **0.0387** | **0.0275** | **0.0315** |

positional information for re-masked tokens, rather than learning molecular representation, which allows the 3D graph encoder to better specialize in MRL.

**Analysis 4. SRD prevents the encoder representation from containing information about 3D coordinates.** To examine whether the encoder captures detailed 3D coordinate information, we train an MLP probe to predict masked atom coordinates from the encoder's representations. We compare the reconstruction loss for encoders pretrained with and without SRD. As shown in Figure 6, the reconstruction loss for the encoder pretrained with SRD is much higher, suggesting that SRD suppresses direct encoding of 3D coordinate details. This forces the encoder to focus on learning higher-level molecular representations that are better aligned with downstream tasks.

**Analysis 5. 2D-PE produces 2D structural context without introducing information leakage.** To assess whether 2D-PE introduces additional information beyond what the 3D graph encoder already captures, we train an MLP to reconstruct the 2D-PE representation from the 3D graph encoder representation and compute the reconstruction error, measured by the cosine similarity between the two representations. During this process, both the 3D graph encoder and 2D-PE are frozen, and only the MLP is trainable. Figure 7 shows that the cosine similarity is very close to 1.0, indicating that the context 2D-PE provided is mostly contained by the 3D graph encoder.

**Analysis 6. 2D-PE encodes structural information for decoding.** We probe whether the pretrained 2D-PE captures structural information required by the decoder. Specifically, we freeze the 2D-PE and train two MLP classifiers to predict atom and bond types from its outputs. Both tasks achieve prediction accuracy above 99.99%, demonstrating that the 2D-PE indeed encodes the structural information necessary for decoding.

# 6 Experiments

## 6.1 Experimental Setup

**Datasets.** For pretraining, we use a large-scale molecular dataset PCQM4Mv2 [55], which contains approximately 3.37 million equilibrium 3D molecular graph structures. For downstream tasks, we evaluate our model on two widely used molecular property prediction datasets: QM9 [13] and MD17 [26]. Specifically, QM9 is a quantum chemistry dataset comprising 134k small molecules, each with its equilibrium conformation and 12 molecular properties (*e.g.,* homo, lumo, dipole moment *etc.*) calculated using density functional theory (DFT). Following prior works [30, 31], we split

Table 2: Performance (MAE ↓) on MD17 force prediction. The best results are **bold**. The second-best results are underline. Results marked with * are reproduced by us.

| Models | Aspirin | Benzene | Ethanol | Malonaldehyde | Naphthalene | Salicylic | Toluene | Uracil |
|---|---|---|---|---|---|---|---|---|
| TorchMD-NET | 0.1216 | 0.1479 | 0.0492 | 0.0695 | 0.0390 | 0.0655 | 0.0393 | 0.0484 |
| 3D-EMGP | 0.1560 | 0.1648 | 0.0389 | 0.0737 | 0.0829 | 0.1187 | 0.0619 | 0.0773 |
| 3D-EMGP (TorchMD-NET) | 0.1124 | 0.1417 | 0.0445 | 0.0618 | 0.0352 | 0.0586 | 0.0385 | 0.0477 |
| Frad* | 0.0825 | **0.1355** | 0.0432 | 0.0535 | 0.0431 | 0.0569 | 0.0433 | 0.0482 |
| 3D-ReTrans | 0.0726 | 0.1619 | 0.0556 | 0.0659 | 0.0423 | 0.0523 | 0.0417 | 0.0427 |
| 3D-GSRD | **0.0583** | 0.1435 | **0.0355** | **0.0468** | **0.0266** | **0.0356** | **0.0274** | **0.0292** |

Table 3: Performance (MAE ↓) on QM9. The best results are **bold**. The second-best results are underline. Results marked with * are reproduced by us.

| Models | $\mu$ (D) | $\alpha$ ($a_0^3$) | $homo$ (meV) | $lumo$ (meV) | $gap$ (meV) | $<R^2>$ ($a_0^2$) | ZPVE (meV) | $U_0$ (meV) | $U$ (meV) | $H$ (meV) | $G$ (meV) | $C_v$ ($\frac{cal}{molK}$) |
|---|---|---|---|---|---|---|---|---|---|---|---|---|
| Uni-Mol2 | 0.089 | 0.305 | - | - | - | 5.26 | - | - | - | - | - | 0.144 |
| SchNet | 0.033 | 0.235 | 41.0 | 34.0 | 63.0 | 0.07 | 1.70 | 14.00 | 19.00 | 14.00 | 14.00 | 0.033 |
| E(n)-GNN | 0.029 | 0.071 | 29.0 | 25.0 | 48.0 | 0.11 | 1.55 | 11.00 | 12.00 | 12.00 | 12.00 | 0.031 |
| DimeNet++ | 0.030 | 0.043 | 24.6 | 19.5 | 32.6 | 0.33 | 1.21 | 6.32 | 6.28 | 6.53 | 7.56 | 0.023 |
| PaiNN | 0.012 | 0.045 | 27.6 | 20.4 | 45.7 | 0.07 | 1.28 | 5.85 | 5.83 | 5.98 | 7.35 | 0.024 |
| SphereNet | 0.025 | 0.045 | 22.8 | 18.9 | 31.1 | 0.27 | **1.12** | 6.26 | 6.36 | 6.33 | 7.78 | 0.022 |
| ComENet | 0.025 | 0.045 | 23.1 | 19.8 | 32.4 | 0.259 | 1.20 | 6.59 | 6.82 | 6.86 | 7.98 | 0.024 |
| TorchMD-NET | 0.011 | 0.059 | 20.3 | 18.6 | 36.1 | **0.033** | 1.84 | 6.15 | 6.38 | 6.16 | 7.62 | 0.026 |
| 3D-ReTrans | 0.016 | 0.055 | 22.0 | 17.8 | 38.0 | 0.341 | 1.85 | 6.18 | 6.36 | 6.51 | 7.89 | 0.029 |
| Transformer-M | 0.037 | 0.041 | 17.5 | 16.2 | **27.4** | 0.075 | 1.18 | 9.37 | 9.41 | 9.39 | 9.63 | 0.022 |
| SE(3)-DDM | 0.015 | 0.046 | 23.5 | 19.5 | 40.2 | 0.122 | 1.31 | 6.92 | 6.99 | 7.09 | 7.65 | 0.024 |
| 3D-EMGP | 0.020 | 0.057 | 21.3 | 18.2 | 37.1 | 0.092 | 1.38 | 8.60 | 8.60 | 8.70 | 9.30 | 0.026 |
| Coord | 0.016 | 0.052 | 17.7 | 14.7 | 31.8 | 0.450 | 1.71 | 6.57 | 6.11 | 6.45 | 6.91 | **0.020** |
| Frad* | 0.012 | 0.045 | **15.4** | **13.7** | 30.6 | 0.428 | 1.56 | 15.88 | 14.67 | 14.87 | 13.52 | 0.023 |
| SliDe* | 0.015 | 0.050 | 18.7 | 16.2 | 28.8 | 0.606 | 1.78 | 10.05 | 10.79 | 11.34 | 11.80 | 0.025 |
| Mol-AE* | 0.152 | 0.434 | - | - | - | 6.962 | - | - | - | - | - | 0.215 |
| Uni-GEM | 0.019 | 0.060 | 20.9 | 16.7 | 34.5 | - | - | - | - | - | - | 0.023 |
| 3D-GSRD | **0.009** | **0.038** | 18.0 | 14.5 | 31.1 | 0.047 | 1.38 | **5.48** | **5.67** | **5.84** | 6.90 | **0.020** |

the dataset into 11,000/1,000/10,831 molecules for training, validation, and testing, respectively. MD17 provides simulated dynamical trajectories for 8 small molecules, including their energy, forces, and conformations. During finetuning, our model first predicts the molecular energy and subsequently derives the forces using the relationship $F = -\nabla_r E$, where $r$ represents the 3D coordinates. For finetuning, we split the dataset into 9500/950 samples for training and validation, and use the remaining samples for testing.

**Baselines.** To evaluate the effectiveness of our proposed framework, we adopt state-of-the-art 3D molecular pretraining methods and supervised models for molecular property prediction as baselines. For 3D molecular pretraining methods, we include Transformer-M [52], SE(3)-DDM [56], 3D-EMGP [57], Coordinate Denoising [32], Fractional Denoising [30], Sliced Denoising [31], UniGEM [58], Mol-AE [19]. For supervised models, we include SchNet [59], E(n)-GNN [60], DimeNet [61], DimeNet++ [62], PaiNN [23], SphereNet [63], TorchMD-NET [24], Uni-Mol2 [16], ComENet [64]. We also include the results of training our backbone (*i.e.,* 3D-ReTrans) from scratch to evaluate the effectiveness of our pretraining methods. We reproduce the results for Frad, SliDe and Mol-AE, while the results for other baselines are directly taken from the referenced papers. More details about baselines and implementation are provided in Appendix D.

## 6.2 Results on MD17

The MD17 dataset contains diverse non-equilibrium molecular structures that are highly sensitive to geometry, making it a challenging benchmark for 3D MRL. As shown in Table 2, 3D-ReTrans achieves performance comparable to TorchMD-NET and surpasses 3D-EMGP on 7 of 8 molecules, demonstrating its strength as a 3D graph encoder for 3D MGM. Moreover, 3D-GSRD attains state-of-the-art results on 7 of 8 molecules except Benzene, exceeding the strongest baseline (*i.e.,* Frad) by a large margin. These results confirm the effectiveness of our pretraining method.

Table 4: Ablation on 3D-ReTrans components. Performance (MAE ↓) on QM9.

| Model Components | homo | lumo | zpve |
|---|---|---|---|
| Relational-Transformer | 27.7 | 24.0 | 1.97 |
| + 3D Data Augmentation | 24.6 | 23.2 | 1.92 |
| + 3D Relational-Attention | 23.4 | 20.3 | 1.90 |
| + 3D Update Layer (3D-ReTrans) | **22.0** | **17.8** | **1.85** |

Table 5: Analyzing SRD on the Relational-Transformer. Performance (MAE ↓) on MD17.

| Decoder | SRD | $\mathcal{L}_{\text{distill}}$ | Toluene | Uracil |
|---|---|---|---|---|
| Structure-dependent | ✓ | ✓ | 0.1144 | 0.0813 |
| Structure-independent | ✗ | ✗ | 0.1250 | 0.0828 |
| Structure-independent | ✓ | ✗ | 0.0998 | 0.0843 |
| Structure-independent | ✓ | ✓ | **0.0745** | **0.0733** |

## 6.3 Results on QM9

We also evaluate the effectiveness of 3D-ReTrans and our pretraining strategy on the QM9 dataset, as shown in Table 3. 3D-ReTrans achieves performance comparable to TorchMD-NET, validating the effectiveness of our proposed backbone architecture. Moreover, 3D-GSRD sets a new state-of-the-art on 7 out of 12 properties, surpassing most baselines, including methods with and without pretraining. These results demonstrate that 3D-GSRD is a highly effective pretraining strategy for MRL, offering advantages over coordinate denoising based approaches.

## 6.4 Ablation Studies and Analysis

**Ablation on Backbone.** To assess the effectiveness of our improvements to the Relational-Transformer [25], we perform ablation studies on each component of 3D-ReTrans, as summarized in Table 4. The results show that 3D Relational-Attention, the 3D Update Layer, and 3D data augmentation each enhance molecular property prediction, collectively boosting overall performance.

**Generalization of SRD Across 3D Graph Encoders.** To evaluate the generalization of SRD, we replace 3D-ReTrans with the Relational-Transformer and conduct additional experiments. As shown in Table 5, incorporating SRD consistently improves downstream performance, demonstrating its effectiveness as a general pretraining strategy applicable to diverse 3D graph encoder architectures.

**Ablation on 2D Graph Position Encoder.** We compare our 2D-PE against alternative structural embeddings, such as those in GraphGPS [37]. Following prior results, we adopt RWSE [65] as a representative baseline due to its strong performance on ZINC and PCQM4Mv2 with relatively low computational cost. As shown in Table 6, replacing 2D-PE with RWSE leads to consistently lower performance, confirming the advantage of 2D-PE in providing 2D structural context.

Table 6: Ablation on 2D graph position encoder. Performance (MAE ↓) on MD17.

| 2D Encoder | Salicylic | Uracil |
|---|---|---|
| RWSE | 0.0368 | 0.0310 |
| 2D-PE | **0.0356** | **0.0292** |

## 7 Conclusion and Future Works

In this work, we introduce 3D-GSRD, a 3D MGM framework with three key components: (1) the Selective Re-mask Decoding that selectively re-masks 3D-relevant information while preserving 2D graph structures; (2) a structure-independent decoder that eliminates all structural information by relying solely on encoder representation; and (3) 3D-ReTrans as the 3D graph encoder for MRL. Our detailed analysis reveals the internal mechanisms of SRD and the structure-independent decoder. Extensive experiments demonstrate that 3D-GSRD significantly outperforms baselines on downstream datasets such as QM9 and MD17.

Despite promising results, several limitations remain. Our pretraining is conducted on PCQM4Mv2 [55] with 3.37M molecules, which is smaller than large-scale datasets such as PubChemQC [66] with 230M molecules, potentially constraining performance. Scaling to larger and more diverse datasets is an important direction. In addition, we focus on molecular property prediction, while other tasks like 3D molecule generation [67–70] and multi-modal molecule–text modeling [71–74] could also benefit from our pretrained autoencoder. Beyond molecular applications, our pretraining paradigm can be extended to broader biological modalities such as single-cell [75] and protein [76].

## Acknowledgement

This research is supported by the National Natural Science Foundation of China (62572449, 624B1012) and National University of Singapore SoC (grant no: A-0010308-00-00).

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

## A  Broader Impacts

This work advances molecular representation learning and has the potential to accelerate downstream applications such as molecular property prediction, drug discovery, molecular dynamics simulation, and material design, helping reduce the cost and time of wet-lab experiments. However, there is a risk of over-reliance on model predictions without sufficient interpretability or domain validation. Additionally, models trained on biased datasets may produce structural or chemical biases, limiting the model's generalization across molecular spaces. We encourage the community to test our models strictly before applying them in scientific scenarios.

## B  More Details on Methodology

In this section, we describe the embedding layer preceding the attention mechanism in 3D-ReTrans. The input 3D molecule graph is defined as $G = (\mathbf{x}, \mathbf{a}, \mathbf{e})$, where $\mathbf{x}$ denotes atomic coordinates, $\mathbf{a}$ represents atom types, and $\mathbf{e}$ denotes pairwise edge features. Our goal is to obtain atomic and edge embeddings that encode both chemical and geometric context.

The initial node embedding $e^{\text{node}}$ jointly encodes atom coordinates and types:

$$e^{\text{node}} = \text{Embed}^{\text{node}}([\mathbf{x}, \mathbf{a}]). \tag{16}$$

To incorporate local geometric context, we compute the neighborhood embedding $e_i^{\text{neigh}}$ for each atom $i$ based on the radial distances $d_{ij}$ to neighbors atom $j$. The radial basis function is given by:

$$e^{\text{RBF}}(d_{ij}) = \phi(d_{ij}) \exp\left(-\beta\left(\exp(-d_{ij}) - \mu\right)^2\right) \tag{17}$$

$$\phi(d_{ij}) = \begin{cases} \frac{1}{2}\left(\cos\left(\frac{\pi d_{ij}}{d_{\text{cut}}}\right) + 1\right), & \text{if } d_{ij} \leq d_{\text{cut}} \\ 0, & \text{if } d_{ij} > d_{\text{cut}} \end{cases} \tag{18}$$

where $\phi(d_{ij})$ is a smooth cutoff function ensuring locality and $\beta$ and $\mu$ are fixed parameters.

The neighborhood embedding $e_i^{\text{neigh}}$ for atom $i$ is then defined as:

$$e_i^{\text{neigh}} = \sum_{j=1}^{N} \text{Embed}^{\text{neigh}}([\mathbf{x}, \mathbf{a}]) \odot \mathbf{W}^r e^{\text{RBF}}(d_{ij}), \tag{19}$$

where $\odot$ denotes element-wise product and $\mathbf{W}^r$ is a learnable projector. We set the cutoff distance $d_{\text{cut}} = 5\,\text{Å}$, ensuring that each atom only attends to neighbors within this spatial range.

The final atomic embedding $e_i^{\text{atomic}}$ combines node and neighborhood information:

$$e_i^{\text{atomic}} = \mathbf{W}^a([e^{\text{node}}, e_i^{\text{neigh}}]), \tag{20}$$

where $\mathbf{W}^a$ denotes learnable projector.

We also obtain the edge embedding $e^{\text{edge}}$ via:

$$e^{\text{edge}} = \text{Embed}^{\text{edge}}(\mathbf{e}). \tag{21}$$

## C  Pseudo Code

We present the pseudocode for pretraining (see Algorithm 1) and finetuning (see Algorithm 2) algorithms in this section.

## D  Experimental Setup

### D.1  Computational Resource

All experiments are conducted on NVIDIA A6000-48G GPUs. Pretraining requires a total of 48 GPU hours. For downstream tasks, finetuning on QM9 and MD17 takes approximately 48 and 8 GPU hours per experiment, respectively.

---
**Algorithm 1** Pretraining of 3D-GSRD
---

**Require:** 3D graph encoder $\phi_\theta^{3D}$, 2D graph encoder $\phi_\theta^{2D}$, decoder $\phi_\theta^{De}$, pretraining dataset $D$, input 3D molecule graph $G = (\mathbf{x}, \mathbf{a}, \mathbf{e})$, masked coordinates prediction head $PosHead_\theta$, denoising prediction head $DenoiseHead_\theta$, mask ratio $p$, denoising loss weight $w$.

1: **while** training is not finished **do**
2:     $G_i = (\mathbf{x}, \mathbf{a}, \mathbf{e}) = $ dataloader$(D)$
3:     randomly mask $p$ atoms and add Gaussian noise ($\Delta x_i \sim 0.04 \cdot \mathcal{N}(0, \sigma^2 I_m)$) to the unmasked atomic coordinates
4:     input moleule $\tilde{G}_i = (\tilde{\mathbf{a}}, \tilde{\mathbf{x}}, \tilde{\mathbf{e}})$
5:     $\mathbf{h_{3D}}, \mathbf{vec} = \phi_\theta^{3D}(\tilde{G}_i)$
6:     $\mathbf{h_{2D}} = \phi_\theta^{2D}(\mathbf{a}, \mathbf{e}))$
7:     $\text{SRD}(\mathbf{h_{3D}}, \tilde{G}_i) = \text{re-mask}(\mathbf{h_{3D}}) + \text{stop-grad}(\mathbf{h_{2D}})$
8:     $\mathbf{rep}, \mathbf{vec} = \phi_\theta^{De}(\text{SRD}(\mathbf{h_{3D}}, \tilde{G}_i), \mathbf{vec})$
9:     For masked atoms: $x_i^{pred} = PosHead_\theta(\mathbf{rep}, \mathbf{vec})$
10:     For unmasked atoms: $\Delta x_i^{pred} = DenoiseHead_\theta(\mathbf{rep}, \mathbf{vec})$
11:     Loss $= ||x_i^{pred} - x_i||_2^2 + w \cdot ||\Delta x_i^{pred} - \Delta x_i||_2^2 - CosineSimilarity(\text{stop-grad}(\mathbf{h_{3D}}), \mathbf{h_{2D}})$
12:     Optimise(Loss)
13: **end while**

---
**Algorithm 2** Finetuning of 3D-GSRD
---

**Require:** 3D graph encoder $\phi_\theta^{3D}$, finetuning dataset $D$, input 3D molecule graph $G = (\mathbf{x}, \mathbf{a}, \mathbf{e})$, label prediction head $LabelHead_\theta$.

1: **while** training is not finished **do**
2:     $G_i = (\mathbf{x}, \mathbf{a}, \mathbf{e}), y_i = $ dataloader$(D)$
3:     $\mathbf{h_{3D}}, \mathbf{vec} = \phi_\theta^{3D}(G_i)$
4:     $y_i^{pred} = LabelHead_\theta(\mathbf{h_{3D}}, \mathbf{vec})$
5:     Loss $= ||y_i^{pred} - y_i||_2^2$
6:     Optimise(Loss)
7: **end while**

---

## D.2 Baselines

We describe the details of our reported baseline methods in this section.

**SchNet** [59] proposes continuous-filter convolutional layers, which enables the model to capture local correlations in molecules without grid-based data.

**E(n)-GNN** [60] introduces an architecture that is equivariant to rotations, translations, reflections, and permutations. Notably, its equivariance extends to higher dimensions with affordable computational overhead increase.

**DimeNet** [61] applies directional message passing which enables graph neural networks to incorporate directional information for molecular predictions and using spherical Bessel functions and spherical harmonics to representation distances and angles.

**DimeNet++** [62] improves upon DimeNet by being 8× faster and achieving 10% higher accuracy, while maintaining strong generalization across molecular configurations and compositions.

**PaiNN** [23] addresses the limitations of invariant representations in message passing neural networks by extending the message passing framework to rotationally equivariant representations.

**SphereNet** [63] analyzes 3D molecular graphs in the spherical coordinate system and propose the spherical message passing (SMP) scheme to efficiently distinguish molecular structures while reducing training complexity.

**TorchMD-NET** [24] introduces an equivariant Transformer architecture with a modified attention mechanism that incorporates interatomic distances directly into the attention weights.

**Transformer-M** [52] is a Transformer based architecture that that handles multiple molecular data modalities within a unified model by using two separate channels to encode 2D and 3D molecular structures.

**SE(3)-DDM** [56] leverages an SE(3)-invariant score matching method to transform coordinate denoising into denoising the pairwise atomic distances within a molecule.

**3D-EMGP** [57] introduces an equivariant energy-based model and develops a self-supervised pre-training framework including a physics-inspired node-level force prediction task and a graph-level noise scale prediction task.

**Coord** [32] proposes a pretraining technique based on denoising for 3D molecular structures, showing it is equivalent to learning a force field.

**Frad** [30] introduces a new hybrid noise strategy by first adding Gaussian noise to the dihedral angles of the rotatable bonds, followed by traditional noise to the atom coordinates, with pretraining focused solely on denoising the latter.

**SliDe** [31] develops a novel sliced denoising method that adds Gaussian noise to bond lengths, angles, and torsion angles with their variances determined by parameters within the energy function.

**Uni-Mol2** [16] is a molecular pretraining model that uses a two-track transformer to jointly capture atomic-level, graph-level, and geometry-level features, while systematically investigating scaling laws in molecular pretraining.

**ComENet** [64] introduces a graph neural network for 3D molecular graphs that adopts rotation angles and local completeness in the 1-hop neighborhood, while integrating quantum-inspired basis functions into its message passing mechanism.

**Mol-AE** [19] addresses the gap between pretraining and downstream objectives in encoder-only 3D molecular models by introducing an auto-encoder with positional encodings as atomic identifiers and a 3D Cloze Test objective that drops atoms to better capture real substructures.

**Uni-GEM** [58] unifies molecular generation and property prediction through a diffusion-based two-phase process of scaffold nucleation and molecule growth, using a multi-branch network with oversampling to balance tasks.

### D.3 Implementation details

We employ the 3D-ReTrans as the 3D graph encoder and implement 2D-PE as 2D-ReTrans, a simplified version of the 3D-ReTrans that excludes 3D coordinates and distance inputs, while using the Transformer as the structure-independent decoder. The 3D graph encoder is configured with a hidden dimension of 256, 8 attention heads, and 12 layers. The 2D-PE shares most of its configuration with the 3D graph encoder, except for a hidden dimension of 64 and 4 attention heads. The decoder consists of 2 layers, with the hidden dimension and number of attention heads same as the 3D graph encoder. The detailed hyper-parameters configuration for pretraining and finetuning are shown in Table 7, Table 8, and Table 9, respectively.

## E More Experimental Results

### E.1 Ablation on Alternative Approach to Eliminating 2D Leakage

We investigate an alternative approach to fully eliminate 2D leakage by directly combining 3D embeddings with 2D positional encodings as molecular representations during downstream tasks. We test this setup with SRD both with and without distillation. As shown in Table 10, simply fusing 2D-PE with 3D-ReTrans under SRD without distillation consistently underperforms our original design, highlighting the necessity of SRD with distillation for effective 2D and 3D alignment. In contrast, when distillation is applied, downstream performance becomes insensitive to whether 2D-PE is explicitly included, suggesting that pretraining distillation sufficiently aligns the modalities and renders additional 2D input unnecessary.

Table 7: Hyper-parameters for pretraining on PCQM4MV2.

| Parameter | Value |
| --- | --- |
| Dataset | PCQM4MV2 |
| Train/Val/Test Split | Others/100/100 |
| Batch size | 128 |
| Inference Batch size | 128 |
| Accumulate grad batches | 2 |
| Optimizer | AdamW |
| Weight decay | 1e-16 |
| Scheduler | CosineAnnealingLR |
| Init learning rate | 5e-5 |
| Min learning rate | 1e-6 |
| Warm up steps | 10000 |
| Max epochs | 30 |
| Masked ratio | 0.25 |
| Masked coordinates reconstruction loss type | MSE loss |
| Coordinate noise scale(type: Gaussian) | 0.04 |
| Denoising loss weight | 0.1 |

Table 8: Hyper-parameters for finetuning on QM9.

| Parameter | Value |
| --- | --- |
| Dataset | QM9 |
| Train/Val/Test Split | 11000/1000/10831 |
| Batch size | 128 |
| Inference Batch size | 128 |
| Accumulate grad batches | 1 |
| Optimizer | AdamW |
| Weight decay | 1e-16 |
| Scheduler | CosineAnnealingLR |
| Init learning rate | 5e-4 |
| Min learning rate | 1e-6 |
| Warm up steps | 1000 |
| Learning rate cosine length | 2,000,000 |
| Max steps | 2,000,000 |
| Max epochs | 2000 |
| Finetuning loss type | MSE loss |

## E.2 Evolution of 2D-PE's representations

To examine how 2D-PE's representation evolves during pretraining, we track the cosine similarity between 2D and 3D representations. As shown in Figure 8, the similarity increases sharply in the initial training steps, approaching 1.0, and then grows gradually, indicating progressive alignment between the two modalities.

Table 9: Hyper-parameters for finetuning on MD17.

| Parameter | Value |
|---|---|
| Dataset | MD17 |
| Train/Val Split | 9500/500/remaining data |
| Batch size | 80 |
| Inference batch size | 64 |
| Accumulate grad batches | 1 |
| Optimizer | AdamW |
| Weight decay | 0.0 |
| Scheduler | CosineAnnealingLR |
| Init learning rate | 5e-4 |
| Min learning rate | 1e-6 |
| Warm up steps | 1000 |
| Max epochs | 1200 |
| Force weight | 0.8 |
| Energy weight | 0.2 |
| Finetuning loss type | MAE loss |
| Ema alpha dy | 1.0 |
| Ema alpha y | 0.05 |

Table 10: Ablation on alternative approach to eliminate 2D leakage. Performance (MAE $\downarrow$) on MD17.

| Downstream Model | SRD | $\mathcal{L}_{\text{distill}}$ | Salicylic | Toluene | Uracil |
|---|---|---|---|---|---|
| 2D-PE+3D-ReTrans | ✓ | ✗ | 0.0420 | 0.0293 | 0.0334 |
| 3D-ReTrans | ✓ | ✗ | 0.0416 | 0.0293 | 0.0329 |
| 2D-PE+3D-ReTrans | ✓ | ✓ | 0.0384 | 0.0275 | 0.0311 |
| 3D-ReTrans | ✓ | ✓ | 0.0387 | 0.0275 | 0.0315 |

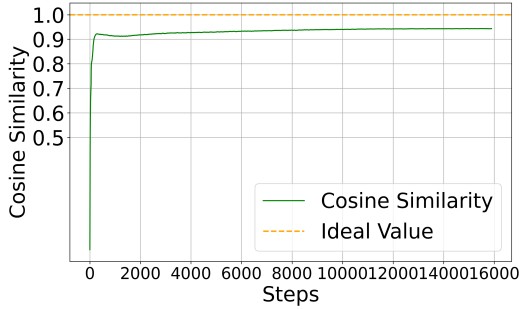

Figure 8: Evolution of the 2D-PE's representation.

