# OpenReview forum: "3D-GSRD: 3D Molecular Graph Auto-Encoder with Selective Re-mask Decoding"
_NeurIPS.cc/2025/Conference — NeurIPS 2025 poster_

### Official Review · Reviewer_Pyfo · 2025-06-08

**Clarity:** 2
**Significance:** 3
**Originality:** 2
**Rating:** 4
**Confidence:** 4

**Summary:**

The paper addresses a key concern in masked graph models: an overly expressive decoder can undermine the encoder, leading to suboptimal representations for downstream tasks. Thus, enhancing reconstruction quality under a restricted decoder is crucial.

**Questions:**

See above.

**Ethical Concerns:**

["NO or VERY MINOR ethics concerns only"]

**Final Justification:**

Thank you for the authors' efforts during the rebuttal period, including the additional experiments. This paper investigates the often-overlooked capabilities of the decoder in Masked Graph Models, providing valuable insights for the community.

**Limitations:**

Yes

**Quality:**

2

**Strengths And Weaknesses:**

Pros:

1.The empirical analysis in Section 5 is well-executed and informative.

2.The authors have released the code, promoting transparency and reproducibility.

3.Investigating potential issues with decoders in masked graph modeling (MGM) is a valuable and timely research direction.

Cons:

1.The method description lacks clear logic and motivation, making it difficult to understand the rationale behind the design choices.

2.It remains unclear whether the 2D-to-3D distillation is intended to transfer knowledge from 3D to 2D. The authors claim this prevents structural leakage in the 2D encoder. However, as stated in the introduction, GNNs inherently reveal structural information. Thus, the motivation for distillation is ambiguous, and the distilled knowledge appears to implicitly encode structural information.

3.Figure 3 is unclear and unintuitive—at first glance, it is not obvious that SRD represents a distillation mechanism. Is it trained jointly with the reconstruction module? This should be clarified.

4.In the related work section, classifying GraphMVP and 3D InfoMax as 2D molecular graph pretraining methods may be inaccurate, as both are fundamentally 3D-based approaches.

5.The two challenges outlined in the introduction have already been discussed in prior work, such as SimSGT. It is unclear what novel perspective or contribution this paper offers in comparison.

6.The baseline methods used in the experiments are somewhat outdated, with most comparisons limited to works from around 2023.

7.The paper appears to introduce a new 3D encoder, 3D-ReTrans, which contradicts the claim of encoder-agnosticism. What happens if existing encoders are used instead? How does this encoder differ from prior works? Additionally, the reported performance differs from results in the original papers:

[1] Spherical Message Passing for 3D Molecular Graphs

[2] ComENet: Towards Complete and Efficient Message Passing for 3D Molecular Graphs

[3] A New Perspective on Building Efficient and Expressive 3D Equivariant Graph Neural Networks

---

> ### Author Rebuttal · Authors · 2025-07-31
>
> > Q1: The method description lacks clear logic and motivation, making it difficult to understand the rationale behind the design choices.
>
> Thank you for the question. We would like to elaborate on the core logic and motivation behind our core design **Selective Re-mask Decoding(SRD)**:
>
> **Goal:** We aim to adapt re-mask decoding for 3D masked graph modeling to better align the reconstruction objective with representation learning (see Lines 34–44 in the introduction).
>
> **Core Observation:** Existing methods face a dilemma:
> * To improve reconstruction quality, some models leak ground-truth 2D structure to the decoder. This harms the encoder’s ability to learn meaningful representations.
> * Others avoid structure leakage by using a structure-agnostic decoder, but this makes reconstruction under re-mask decoding very challenging—since the decoder cannot infer the relative positions of multiple masked atoms.
>
> To formalize this issue, we define:
> * **Definition (Structure Leakage in Masked Graph Auto-encoders).** In a masked graph auto-encoder, structure leakage refers to the use of any 2D structural information (e.g., adjacency matrix) as input to the graph decoder, unless it is conveyed solely through the graph encoder’s output. Accessing structure only via the graph encoder's output does not constitute structure leakage.
> * _**Definition Explanation.**_ If the decoder has access to structure outside the encoder’s output, it can ignore the encoder, undermining the encoder's ability to learn useful representations.
>
> **Our Idea:** To resolve the challenge, we propose SRD:
> * Applies standard re-masking to atom representations;
> * Add the re-masked representations with a 2D positional encoding (PE);
> * Crucially, this 2D PE is trained via distillation to mimic the 3D graph encoder’s output. In this way, the 2D PE does not provide information beyond the 3D encoder's output, thus avoiding structure leakage.
>
> We have revised the method section to better emphasize this logic, and we hope this clarifies our motivation and design choices.
>
> > Q2: The motivation for distillation is ambiguous, and the distilled knowledge appears to implicitly encode structural information.
>
> To answer the question, we explain that **the 2D PE can provide structural information, but are not structure leakage because these information are mostly contained in the 3D encoder.**
>
> * **2D-to-3D distillation ensures that the 2D PE's embedding are mostly contained by the 3D encoder's output.** 2D-PE's representations are supervised by 3D encoder outputs through distillation. In Analysis 5, we empirically show that the 2D-PE's representations can be mostly recovered from the 3D encoder's output (nearly 1.0 cosine similarity).
> * **By definition, structure information provided by the 3D encoder is not leakage.** Please refer to our response to W1 for the definition of structure leakage in masked graph auto-encoders. Considering the embedding provided by the 2D-PE is mostly contained in the 3D encoder's output, they are not structure leakage.
>
> > Q3: Figure 3 is unclear and unintuitive—at first glance, it is not obvious that SRD represents a distillation mechanism. Is it trained jointly with the reconstruction module? This should be clarified.
>
> To clarify, Selective Re-mask Decoding (SRD) includes two components that work together during training:
> * **Re-mask decoding with 2D PE.** The decoder is trained to reconstruct masked atom positions using both the 3D encoder output and 2D-PE representations. The 2D-PE provide necessary 2D structural context, enabling the decoder to distinguish the relative positions of re-masked atoms for better reconstruction.
> * **2D-from-3D distillation.** This is a joint loss function with the reconstruction module, aiming to train the 2D-PE. Specifically, 2D-PE is trained via knowledge distillation from 3D encoder, using cosine similarity loss over unmasked atoms. The stop-gradient operation can enforce 2D-PE to foucus on learning from 3D encoder's representation and 3D encoder to specialize in representation learning, respectively. The total pretraining loss is the sum of the MSE loss and the consine similarity loss.
> We have revised our Section 4.1 and Figure 3 to make this clearer.
>
> > Q4: In the related work section, classifying GraphMVP and 3D InfoMax as 2D molecular graph pretraining methods may be inaccurate, as both are fundamentally 3D-based approaches.
>
> We initially categorized GraphMVP and 3D InfoMax as 2D molecular graph pretraining methods because they focus on enhancing 2D encoders using 3D information and apply 2D graphs in downstream tasks. However, we agree it is more accurate to classify them as 3D pretraining methods due to their reliance on 3D signals during training. We have updated the related work section accordingly.
>
> > Q5: The two challenges outlined in the introduction have already been discussed in prior work, such as SimSGT. It is unclear what novel perspective or contribution this paper offers in comparison.
>
> We respectfully clarify that the two challenges introduced in our paper are fundamentally distinct from those addressed in SimSGT:
> * **Challenge 1** is about the 2D structure leakage in the graph decoder, a phenomenon that is neither discussed nor mitigated in SimSGT.
> * **Challenge 2** concerns reconstructing masked graphs without introducing 2D structure leakage—a capability not explored in SimSGT.
>
> Beyond these conceptual differences, our work also introduces several key technical novelties and differences:
>
> - **Selective Re-mask Decoding (SRD):** Our SRD strategy explicitly prevents 2D structure leakage by controlling the flow of 2D information into the graph decoder. This contrasts with SimSGT’s re-masking scheme, which still allows such leakage.
> *	**3D-ReTrans Architecture:** We propose a novel 3D graph encoder-decoder architecture designed for 3D masked graph modeling. This architectural design is not present in SimSGT or other prior work.
>
> **Different Problem Scopes:**  Our study focuses on 3D masked graph modeling, whereas SimSGT addresses only the 2D setting. This difference in scope further differentiates the contributions of our work.
>
> In summary, both the challenges we define and the solutions we propose are orthogonal to those in SimSGT, and collectively constitute a novel contribution to the field.
>
>
> > Q6: The baseline methods used in the experiments are somewhat outdated, with most comparisons limited to works from around 2023.
>
> In response, we have added a recent 2025 baseline UniGEM[1] in our revised manuscript to ensure a fairer comparison. As shown in Table 1, our method outperforms UniGEM across 6 tasks in QM9, further validating its effectiveness.
>
> |Model| α | homo | lumo  | gap| μ  |Cv
> |---|---|---|---|---|---|---
> |UniGEM| 0.060 | 20.9 |16.7| 34.5| 0.019| 0.023
> |3D-GSRD | **0.038** |**18.0** | **14.5** |**31.1** |**0.009**|**0.020**
>
> **Table 1: Results on QM9 dataset.**
>
> > Q7.1: The paper appears to introduce a new 3D encoder, 3D-ReTrans, which contradicts the claim of encoder-agnosticism. What happens if existing encoders are used instead?
>
> We introduced 3D-ReTrans because it achieved the best performance in our experiments, but our core contribution, the SRD mechanism, is itself encoder-agnostic. To verify the encoder-agnostic nature of our method, we conducted additional experiments using the Relational Transformer in place of our proposed 3D-ReTrans. The results, presented in Table 2, demonstrate that adding SRD  consistently improves performance. This shows that SRD, as a general pre-training strategy, can empower different 3D encoder architectures.
>
> |Decoder| SRD | L_distill |Toluene  | Uracil
> |---|---|---|---|---
> |Structure-dependent |√  |√  |0.1144 | 0.0813
> |Structure-independent | × |× |0.1250| 0.0828
> |Structure-independent |√  |× |0.0998 | 0.0843
> |Structure-independent | √ | √ |0.0745 | 0.0733
>
> **Table 2: Results on Relational-Transformer encoder.**
>
> > Q7.2: How does this encoder differ from prior works?
>
> Here we summarize the key innovations and differences of our encoder over several major prior works:
>
> **Relational Attention.** Unlike standard Relational Attention, which directly injects pair features into attention weights, we separate and jointly process scalar features (e.g., atom types and distances) and vector features (e.g., directional geometry). Our 3D Relational-Attention captures pairwise interactions and integrates them into both scalar and vector channels, followed by a 3D Update Layer that enables interaction between scalar and vector features.
>
> **TorchMD-Net.** TorchMD-Net's attention mechanism primarily include interatomic distances. We go further by incorporating edge vectors into attention. These are concatenated with node vectors before QKV projection.
>
> **Uni-Mol.** Uni-Mol uses separate atom and pair representations, connected via static attention bias. In contrast, we dynamically process scalar and vector features with cross-channel interaction, enabling more expressive modeling of molecular geometry and structure.
>
> > Q7.3: Additionally, the reported performance differs from results in the original papers:
>
> The SphereNet [2] results reported in our paper follow those provided by Frad [5]. We acknowledge that these may differ from the original paper due to differences in reproduction settings. To address this, we have updated our manuscript to include the original SphereNet results from [2]. Additionally, we appreciate your suggestion and will include [3] and [4] as additional baselines in the final version to ensure a more comprehensive comparison.
>
> [1] UniGEM: A Unified Approach to Generation and Property Prediction for Molecules. ICLR 2025
>
> [2] Spherical message passing for 3d molecular graphs
>
> [3] ComENet: Towards complete and efficient message passing for 3D molecular graphs
>
> [4] A new perspective on building efficient and expressive 3D equivariant graph neural networks
>
> [5] Fractional denoising for 3d molecular pre-training

---

> > ### Comment · Reviewer_Pyfo · 2025-08-02
> > **Thank You**
> >
> > Thank you for your rebuttal, My main concern are resolved.

---

> > > ### Author Response · Authors · 2025-08-02
> > > **Thanks for the feedback**
> > >
> > > Thank you very much for acknowledging our rebuttal and efforts. We're glad that our rebuttal helped address your concerns. Your feedback has been very helpful in improving our work, and we will incorporate the corresponding refinements in the final version of the paper.

---

### Official Review · Reviewer_Q2ph · 2025-06-25

**Clarity:** 3
**Significance:** 3
**Originality:** 3
**Rating:** 5
**Confidence:** 4

**Summary:**

This paper introduces a novel masked modeling approach for 3D molecular pre-training, together with a dedicated molecule transformer architecture designed to complement this pre-training method. The central contribution is a 3D molecular autoencoder pre-training scheme that addresses the issue of 2D structural information “leakage” to the molecule decoder, which has limited the effectiveness of previous masked molecule modeling frameworks. To tackle this, the authors decouple the 2D graph encoder from the 3D graph encoder, providing masked atoms with 2D structure information exclusively through a separate encoder. This 2D encoder is aligned with the 3D encoder via contrastive learning, which both distills 3D information into the 2D representation and enforces consistency between the modalities. In addition, the paper proposes a new molecule transformer architecture for 3D encoding, inspired by TorchMD-Net but enhanced with 3D data augmentation strategies to promote equivariant learning—an approach that is, to my knowledge, novel in this context.

**Questions:**

1. Can you report or discuss the computational efficiency of the proposed masked modeling method and the new transformer architecture, in comparison to prior approaches?
2. A straightforward baseline for the 2D encoder would be to use structural embeddings, as in the GraphGPS paper. Could the authors provide an ablation study comparing their contrastive alignment strategy to such alternatives?

**Ethical Concerns:**

["NO or VERY MINOR ethics concerns only"]

**Final Justification:**

I appreciate the authors' responses to my concerns. The additional experiments validating the 2D encoder's capture of structural information effectively address my main concern. The comparison with GraphMVP is also clear, highlighting both similarities and key differences in distillation targets and loss designs. The ablation studies quantify the contributions of innovations and 3D data augmentations. My previous concerns have been addressed with additional experiments and clear explanations, and I have raised my score to 5.

**Limitations:**

yes

**Paper Formatting Concerns:**

No major formatting issues were identified.

**Quality:**

3

**Strengths And Weaknesses:**

**Strengths**

1. The authors cleverly noticed the 2D graph structure leakage to the molecule decoder, which is a rather novel observation. Addressing this issue, they used a separate 2D graph encoder to introduce 2D structure information, and apply a structure-unaware molecule decoder. This 2D graph encoder is trained by contrastive learning with the 3D encoder. Overall, I feel the observed challenge is insightful, and the proposed solution is intricate but makes sense.
2. The method is empirically validated with strong performance on relevant benchmarks, demonstrating improvements over competitive baselines. The experimental section (Section 5) provides supporting analysis for key design choices.
3. The paper is generally well-written. The technical motivation, architectural decisions, and experimental results are clearly articulated, making the work accessible to the community.


**Weakness**

1. The paper would benefit from more detailed analysis and visualization of the learned 2D encoder. For example, does the 2D encoder capture the necessary structural information required by the decoder, and how does this representation evolve during contrastive training?
2. The 2D-from-3D distillation approach bears some resemblance to the methodology in GraphMVP. A clearer discussion of similarities, differences, and relative advantages compared to GraphMVP would strengthen the paper.
3. While the proposed molecule transformer is motivated as an advance over relational transformers, its specific improvements over TorchMD-Net are less clear. Further ablation studies isolating the contributions of the new architectural elements (e.g., 3D data augmentations) would be helpful.

---

> ### Author Rebuttal · Authors · 2025-07-31
>
> > Weaknesses1: The paper would benefit from more detailed analysis and visualization of the learned 2D encoder. For example, does the 2D encoder capture the necessary structural information required by the decoder, and how does this representation evolve during contrastive training?
>
> **Response:** Thank you for the valuable suggestions. We provide additional analysis of the 2D encoder:
> (1) To show whether the 2D encoder capture strctural information required by the decoder, we perform probing experiments. Specifically, we freeze our pretrained 2D encoder and train two MLPs to predict atom types and bond types from its output. The resulting prediction accuracy for both atom and bond types exceed 99.99%, indicating that the 2D encoder effectively encodes strutural information needed by decoding.
> (2) To visualize the evolution of the 2D encoder’s representation, we draw the cosine similarty curve between 2D representation and 3D representation during training. The similarity rises sharply within the first few training steps and approaches 1.0, followed by a gradual increase.
> We will include these results and visualizations in the final version to improve the clarity and completeness of our analysis.
>
>
> > Weaknesses2: The 2D-from-3D distillation approach bears some resemblance to the methodology in GraphMVP. A clearer discussion of similarities, differences, and relative advantages compared to GraphMVP would strengthen the paper.
>
> **Response:** Thank you for the insightful comment. We provide a clearer discussion of our 2D-from-3D distillation approach and GraphMVP below:
>
> **Similarity:** Both methods leverage information from 2D and 3D graphs to improve representation learning.
>
> **Differences:**
> - GraphMVP transfers 3D information into a 2D encoder to enhance its downstream performance. In contrast, our method distills 3D representation into 2D-PE, ensuring the 2D-PE's embedding is contained within the 3D encoder to avoid structure leakage in decoding.
> - GraphMVP aligns the 2D and 3D views of the same molecule and contrasts the views of different molecules using contrastive losses such as InfoNCE or EBM-NCE. Our method instead uses a cosine similarity loss that encourages 2D-PE to generate structural encodings closely aligned with the 3D encoder, allowing both to advance jointly during pretraining.
>
> **Relative advantages:** Our 2D-from-3D distillation is simpler and efficient, using only a lightweight cosine similarity loss rather than complex contrastive frameworks method to achieve our goal of 2D structure-informed decoding without structure leakage.
>
> We have revised the related work section to include this discussion.
>
>
> > Weaknesses3: While the proposed molecule transformer is motivated as an advance over relational transformers, its specific improvements over TorchMD-Net are less clear. Further ablation studies isolating the contributions of the new architectural elements (e.g., 3D data augmentations) would be helpful.
>
> **Response:** Thank you for the helpful suggestion. We clarify that our model is not a direct modification of TorchMD-Net. Instead, we borrow its key components (the scalar and vector features) to develop 3D relational-Attention and 3D Update Layer. In addition, unlike TorchMD-Net, which is inherently equivariant, we use 3D data augmentations to encourage 3D-ReTrans to learn 3D equivariance and invariance. This is important because 3D-ReTrans lacks built-in 3D equivariance or invariance.
>
> To validate the contribution of each component, we conduct ablation studies in Table 1. The results demonstrate that both the architectural innovations and the 3D data augmentation significantly contribute to the model's performance.
>
>
> |Model Components |homo|
> |---------------|---------|
> | Relational-Transformer |27.7 |
> | +Data Augmentation  | 24.6 |
> | +3D Relational-Attention |23.4 |
> | +3D Update Layer(3D-ReTrans) | **22.0** |
>
> **Table 1: Ablation of different components of 3D-ReTrans.**
>
>
> > Q1: Can you report or discuss the computational efficiency of the proposed masked modeling method and the new transformer architecture, in comparison to prior approaches?
>
> **Response:** Thank you for the suggestion. We report the computational efficiency of our method and Frad in the table below. All experiments are conducted on NVIDIA A6000 (48GB) GPUs. Since the Frad paper does not provide GPU-hour estimates, we reproduced their method and approximated the training cost under comparable settings. For pretraining, we use a batch size of 128 and train for 30 epochs. For downstream tasks, we follow this setups: 1500 epochs with batch size 128 for QM9, and 1400 epochs with batch size 100 for MD17. As shown in the table, our method achieves competitive training efficiency compared to prior approaches.
>
>
> |GPU Hours| Pretrain | QM9 | MD17  |
> |----|---------|---------|---------|
> | Frad| 55h  |  50h |8h
> | Ours | 48h | 48h | 8h  |
>
> **Table 2: Computational overhead of our method and Frad.**
>
> > Q2: A straightforward baseline for the 2D encoder would be to use structural embeddings, as in the GraphGPS paper. Could the authors provide an ablation study comparing their contrastive alignment strategy to such alternatives?
>
> **Response:**   Thank you for the susggestion. Following GraphGPS, we conduct pretrain and finetune experiments with the RWSE[1] structure encoding to replace our 2D PE's function. For pretraining, we find that pretrain loss cannot converge as low as the original model. For finetuning, our results are shown in Table 3. Our 2D-PE's performance consistently surpasses the setting using RWSE, indicating the superior of 2D-PE to provide effective 2D structural context.
>
> |2D Encoder| Salicylic | Uracil  |
> |----|---------|---------|
> |RWSE  | 0.0371  | 0.0312 |
> |Ours 2D-PE |  **0.0356**| **0.0292**|
>
> **Table 3:  Results on MD17 dataset.**
>
> We also show the limitation of RWSE and other kinds of graph structure embedding below:
> 1. Neglect of chemical bond types. RWSE primarily encodes structural information such as relative distance and direction between nodes using random walk transition probabilities. However, it neglect chemical bond types and atom types.
> 2. Lossy representation. The walk length K in RWSE is fixed and typically cannot be set too large due to computational constraints. This limits the scope of the structural encoding. As a result, RWSE may fail to capture long-range dependencies in complex molecular graphs, leading to incomplete or lossy representations of global molecular structures.
>
>
> [1] Dwivedi V P, Luu A T, Laurent T, et al. Graph neural networks with learnable structural and positional representations[J]. arXiv preprint arXiv:2110.07875, 2021.

---

### Official Review · Reviewer_GqqD · 2025-07-03

**Clarity:** 4
**Significance:** 3
**Originality:** 2
**Rating:** 4
**Confidence:** 4

**Summary:**

This paper addresses a core dilemma in 3D molecular modeling using re-mask decoding: whether or not to leak 2D information into the decoder. The authors propose a clean solution by introducing a student–teacher framework that selectively reveals only the 2D information already learned by the encoder to the decoder. This effectively mitigates the 2D leakage issue. Experimental results on downstream 3D molecular tasks demonstrate the effectiveness of the proposed approach.

**Questions:**

Could the authors conduct an additional experiment based on Table 1? I'm particularly curious about the performance when using SRD without distillation, but instead representing the molecule with 3D embeddings combined with 2D positional encoding for downstream tasks.

It would also be helpful to try this setup on your current with-distillation model as well. Since it doesn’t require redoing pre-training, this experiment should be feasible to complete during the rebuttal period.

**Ethical Concerns:**

["NO or VERY MINOR ethics concerns only"]

**Final Justification:**

The author has already addressed most of my concerns. I still maintain my positive assessment and score.

**Limitations:**

yes

**Quality:**

3

**Strengths And Weaknesses:**

**Strengths**:

1, The paper is well-written, with a clear motivation and compelling empirical evidence (e.g., Figure 2) supporting the proposed approach. And all the key claims made early on are later validated through ablation studies.

2, This paper actually effectively addresses two critical challenges: (1) the pre-training vs. downstream mismatch in 3D molecular modeling, resolved by reusing the re-mask technique from 2D models; and (2) the inherent conflicts in re-mask decoding. Both issues are highly relevant to 3D molecular representation learning, and the proposed methods and insights are likely to be influential for the community.

**Weaknesses**

1, My main concern lies in the magnitude of the technical contribution. While the analysis are strong, the technical novelty appears limited. Although the authors claim to use a graph auto-encoder, the model architecture is in fact very similar to transformer-based models already used in the 3D molecular domain, such as Uni-Mol and Mol-AE, both of which are already cited in the paper. In particular, the design is highly similar to Mol-AE, with almost identical architecture and atom drop strategy — the only difference lies in the positional encoding used in the decoder.

2, The use of a distilled PE does not fully eliminate 2D leakage. For example, in an image–text CLIP procedure, we cannot assume that aligned embeddings are completely free of modality-specific information. Additionally, in this paper’s setting, the information of dropped atoms is not present in the 3D encoder output, so their 2D structure is inevitably leaked to the decoder by 2D PE.

3, There exists a much simpler solution to eliminate 2D leakage altogether. During downstream evaluation, you could directly use the 3D embedding combined with the 2D PE as the final molecular embedding—potentially eliminating the need for the distillation design entirely.

---

> ### Author Rebuttal · Authors · 2025-07-31
>
> > Weaknesses1: My main concern lies in the magnitude of the technical contribution. While the analysis are strong, the technical novelty appears limited. Although the authors claim to use a graph auto-encoder, the model architecture is in fact very similar to transformer-based models already used in the 3D molecular domain, such as Uni-Mol and Mol-AE, both of which are already cited in the paper. In particular, the design is highly similar to Mol-AE, with almost identical architecture and atom drop strategy — the only difference lies in the positional encoding used in the decoder.
> >
>
> **Response:**
> Thank you for your thoughtful comments. We respectfully clarify that our major novelty compared to Mol-AE and UniMol includes: (1) our Selective Re-mask Decoding (SRD) strategy, in which the 2D-PE is a key element; and (2) 3D-ReTrans, which combines the strength of TorchMD-net and Relational Transformer for 3D encoding and 3D masked graph modeling. Here we elaborate their novelties below:
>
> (1) SRD prevents 2D structure leakage in decoder while ensuring 2D structure-informed decoding, which is not addressed in Mol-AE and UniMol. The novelty of SRD lies in:
> * **2D-from-3D Distillation to ensure no structural leakage:** we train 2D-PE via distilled from 3D encoder's representations ensuring its information fully contained within 3D encoder, which is absent in Mol-AE and UniMol.
> * **2D ReTrans-based positional encoding:** we leverage 2D-ReTrans to produce high-quality 2D positional encodings for re-masked decoding. In contrast, Mol-AE relies on SMILES-based positional information, which is less structured and less effective.
>
> (2) 3D-ReTrans outperforms UniMol and Mol-AE, through relational attention and joint scalar and vector features. Both UniMol and Mol-AE employs the Uni-Mol architecture, which can not fully capture interaction between atoms.
>
> **Empirical Validation:** As shown in Table 1, our ablation study on QM9 demonstrates that 3D-ReTrans outperforms both Uni-Mol2 (1.1B) and Mol-AE. The results for Uni-Mol2 are taken directly from [1].
>
> | **Model**     | **$\mu$** | **alpha** | **R2**  | **Cv**  |
> |---------------|-----------|-----------|---------|---------|
> | Uni-Mol2 1.1B |  0.089   | 0.305   | 5.265   | 0.144   |
> | Mol-AE        | 0.152    |0.434  | 6.962   |0.215    |
> | 3D-ReTrans    | 0.016     | 0.055     | 0.341   | 0.029   |
>
> **Table1: Ablation study on QM9.**
>
>
>
> > Weaknesses2: The use of a distilled PE does not fully eliminate 2D leakage. For example, in an image–text CLIP procedure, we cannot assume that aligned embeddings are completely free of modality-specific information. Additionally, in this paper’s setting, the information of dropped atoms is not present in the 3D encoder output, so their 2D structure is inevitably leaked to the decoder by 2D PE.
>
> **Response:**  Thank you for your concern regarding potential structure leakage. The 2D PE provides 2D structure information, but is not structure leakage, because these information are included in the 3D encoder's embedding. We explain this by giving a formal definition for structure leakage:
>
> * **Definition (Structure Leakage in Masked Graph Auto-Encoders).** In a masked graph auto-encoder, structure leakage refers to the use of any 2D structural information (e.g., adjacency matrix) as input to the graph decoder, unless it is conveyed solely through the graph encoder’s output. Accessing structure only via the graph encoder's output does not constitute structure leakage.
> * _**Definition Explanation.**_ If the decoder has access to structure outside the encoder’s output, it can ignore the encoder, undermining its ability to learn useful representations.
>
> **Why 2D-PE does not constitute structure leakage?** It is learned via 2D-from-3D distillation, i.e.,trained to mimic representations derived from the 3D encoder. As shown in Analysis 5 in our paper, the representation produced by 2D-PE is mostly embedded in the 3D encoder's output. Thus, the decoder does not receive any structural information beyond what is already captured by the encoder, satisfying our definition above.
>
>
>
> > Weaknesses3: There exists a much simpler solution to eliminate 2D leakage altogether. During downstream evaluation, you could directly use the 3D embedding combined with the 2D PE as the final molecular embedding—potentially eliminating the need for the distillation design entirely.
>
>
> **Response:** Thank you for the insightful suggestion. Following your advice, we perform experiments on the MD17 dataset, and present the results below:
>
> |Downstream Model| SRD | L_distill |Salicylic| Toluene  | Uracil  |
> |-------------|---------|-----------|---------|---------|---------|
> | 2D-PE+3D-ReTrans |√  |×  |0.0420  | 0.0293 |0.0334|
> | 3D-ReTrans    | √ |× | 0.0416 | 0.0293 |0.0329 |
> | 2D-PE+3D-ReTrans |√  |√ | 0.0384 | 0.0275| 0.0311  |
> | 3D-ReTrans    | √ | √ | 0.0387 | 0.0275 | 0.0315 |
>
> **Table2: Ablation study on MD17 dataset.**
>
> We can observe that using our SRD method with distillation present better performance. This demonstrates the benefit of aligning 2D-PE with the 3D encoder during pretraining.
>
> **Why directly training the 2D PE can diminish the role of the 3D encoder in 3D masked graph modeling?** Jointly training the 2D PE with the 3D graph encoder can indeed allow the model to provide 2D structural context to the re-masked atoms. However, without distillation, this setup tends to decouple the learning of 2D and 3D representations: the 2D-PE tends to specialize in capturing 2D topological information, while the 3D encoder focuses solely on 3D geometric features. During pretraining, this separation is less problematic because the decoder has sufficient capacity to integrate both signals. However, this becomes a limitation in downstream fine-tuning, where the fused representation is typically passed through a lightweight MLP projector. The MLP's limited capacity makes it difficult to effectively combine the independently learned 2D and 3D features.
>
>
> > Q1: Could the authors conduct an additional experiment based on Table 1? I'm particularly curious about the performance when using SRD without distillation, but instead representing the molecule with 3D embeddings combined with 2D positional encoding for downstream tasks.
>
> **Response:** Thank you for the question. As shown in Table 2 above, the performance of using SRD without distillation and combing 2D-PE and 3D-ReTrans for downstream finetuning is consitently lower than our original setup with distillation. This highlights the importance of our proposed SRD strategy in aligning 2D and 3D representations for more effective downstream integration.
>
>
> > Q2: It would also be helpful to try this setup on your current with-distillation model as well. Since it doesn’t require redoing pre-training, this experiment should be feasible to complete during the rebuttal period.
>
> **Response:** As shown in Table 2 above, when distillation is applied, the performance is similar regardless of whether 2D-PE is incorporated during downstream finetuning. This suggests that once the 2D-PE is properly aligned through distillation during pretraining, explicitly including it at the downstream stage offers very limited additional benefit.
>
>
> [1] Ji X, Wang Z, Gao Z, et al. Uni-mol2: Exploring molecular pretraining model at scale[J]. arXiv preprint arXiv:2406.14969, 2024.

---

### Note · Authors · 2025-08-12

Dear Reviewers, ACs, SACs, and PCs,

We sincerely thank you for your time and feedback. The rebuttal process has greatly strengthened our work. Below, we outline our main contributions, the clarifications and experiments added during rebuttal, and our plans for the final version.

**Positive feedback from reviewers:**

- **[Reviewer GqqD, Q2ph, Pyfo] Novelty & Significance:** All reviewers agreed that tackling 2D structural information leakage in 3D molecular masked auto-encoders is important.
- **[Reviewer GqqD, Q2ph] Technical Contribution:** Our Selective Re-mask Decoding was recognized as a sound and meaningful approach.
- **[Reviewer GqqD, Q2ph, Pyfo] Empirical Evidence:** Experiments were considered well-executed, informative, and strongly supportive of our claims.
- **[Reviewer GqqD, Q2ph] Clarity & Presentation:** The paper was generally viewed as well-written with clear motivation and technical details.

**Key improvements during rebuttal:**

- **[Reviewer GqqD] Clarified novelty vs. Mol-AE & Uni-Mol:** We clarified the novelty of our SRD and the 3D-ReTrans compared to existing works, with ablation study showing large gains on QM9.
- **[Reviewer GqqD] Explored alternative to SRD:** We tested a simpler variant combining 3D embeddings and 2D PE, showing the effectiveness of SRD.
- **[Reviewer Q2ph] Baseline for 2D PE:** We compared against GraphGPS structural encodings (RWSE), analyzed their limitations, and showed our method’s superiority.
- **[Reviewer Q2ph] 2D PE Analysis:** We confirmed 2D PE’s ability to capture structural information through probing and cosine similarity analyses.
- **[Reviewer Q2ph] Ablation Study for 3D-ReTrans:** We include ablation study for 3D-ReTrans for homo, lumo and zpve, validating the effectiveness of our architectural design.
- **[Reviewer Pyfo] Clarified motivation & Figure 3:** Clarified our two main challenges, the motivation and logic of SRD design, distillation, and Figure3.
- **[Reviewer Pyfo] Encoder-agnostic validation:** We validated SRD on other encoders ( Relational Transformer) and summarized innovations over prior 3D GNNs.

These efforts resolved the main concerns. Reviewer Q2ph raised score, and Reviewer Pyfo confirmed main issues were resolved.

**Final version commitments:**

- **[Reviewer GqqD, Q2ph, Pyfo]** Add new results and analyses (more ablations, RWSE, other baselines).
- **[Reviewer Q2ph, Pyfo]** Improve clarity by strengthening related work and refining motivation and method logic.

---

### Decision · Program_Chairs · 2025-09-17

**Decision:**

Accept (poster)

**Comment:**

This paper proposes a selective re-mask decoding strategy for 3D molecular representation learning. To this end, the authors focus on solving the 2D structural leakage issue, where the encoder fails to learn meaningful representations due to the decoder being able to fully reconstruct the 3D coordinates from 2D structures.

Reviewers were concerned about the novelty of neural architecture and that the use of distilled 2D position encodings may not completely eliminate structure leakage. The rebuttal have somewhat resolved these concerns.

Overall, while the conceptual advance is modest, the paper is technically sound, demonstrates clear empirical benefits, and provides meaningful insights into the design of 3D molecular pretraining methods. I recommend acceptance for the paper.